# Compromised Cranio-Spinal Suspension in Chiari Malformation Type 1: A Potential Role as Secondary Pathophysiology

**DOI:** 10.3390/jcm11247437

**Published:** 2022-12-15

**Authors:** Belinda Shao, Jonathan A. Poggi, Natalie Amaral-Nieves, Daniel Wojcik, Kevin L. Ma, Owen P. Leary, Petra M. Klinge

**Affiliations:** Department of Neurosurgery, Brown University, Providence, RI 02903, USA

**Keywords:** Chiari Malformation Type I, cisterna magna, myodural bridges, dentate ligaments, craniocervical junction, arachnoid adhesions, spinal cord suspension, viscoelasticity

## Abstract

In Chiari Malformation Type I (CM1), low-lying tonsils obstruct the cisterna magna at the foramen magnum, thereby compromising the essential juncture between the cranial and spinal compartments. The anatomical obstruction of the cisterna magna inhibits bi-directional CSF flow as well as CSF pulse pressure equilibration between the intracranial compartment and the intraspinal compartment in response to instances of increased intracranial pressure. Less understood, however, are the roles of the spinal cord suspension structures at the craniocervical junction which lend viscoelastic support to the spinal cord and tonsils, as well as maintain the anatomical integrity of the cisterna magna and the dura. These include extradural ligaments including the myodural bridges (MDBs), as well as intradural dentate ligaments and the arachnoid framework. We propose that when these elements are disrupted by the cisterna magna obstruction, tonsillar pathology, and altered CSF dynamics, there may arise a secondary pathophysiology of compromised and dysfunctional cranio-spinal suspension in CM1. We present intraoperative images and videos captured during surgical exposure of the craniocervical junction in CM1 to illustrate this proposal.

## 1. Introduction

### 1.1. Intra- and Extradural Cranio-Spinal Suspension and the Cisterna Magna (CM)

The cisterna magna (CM) is the cistern at the craniospinal junction. It has a strategic location inferior to the cerebellum and the medulla and dorsal to the upper cervical spinal cord. Intra-durally, the CM harbors the cerebellar tonsils, the lower cranial nerves, the C1 nerve and dentate ligaments as well as the PICA branches, the latter known to show natural and unpredictable variations. Like all cranial cisterns, the CM allows compensation of acute intracranial and intraspinal pressure changes under both physiologic and pathologic conditions providing a capacious subarachnoid cerebrospinal (CSF) compartment. Extra-durally, the cisterna magna is enclosed by the bony foramen magnum and ring of the C1 vertebral arch, which are attached to the dura by transitional viscoelastic fibrous structures called myodural bridges (MDBs) that extend through these atlanto-occipital and atlanto-axial interspaces [1,2,3,4,5,6]. In their role as a bridging structure, the MDBs also act to suspend the dura from the suboccipital musculature and tent open the cisterna magna [7]. This complex then connects with the rest of the craniocervical junction via the suboccipital musculature. (Figure 1: The suspensive myodural bridge complex at the cisterna magna).

Extradural suspension of the dura mater likely supports and augments this function of the cisterna magna through mechanical forces, driving CSF flow between the cranial and spinal subarachnoid spaces and suspending the thecal sac against the complex movements of the bony atlantoaxial and atlantooccipital joints. Hofman ligaments, or meningovertebral ligaments, are small connective structures in the epidural space that tether the thecal sac to the osseous spinal canal yet allow it to adjust to the many dynamic movements of bony spine. Moreover, during neck flexion, extension, and rotation, the MDBs have been demonstrated to pull on the dural sac with the contraction of suboccipital muscles, and change the volume of CSF in the subarachnoid space, thereby acting as a dynamic pump for CSF across the cisterna magna [1,3,4,8]. Such movements have also been shown to move the tonsils themselves both in people with and without CMI [9].

The intradural suspension of the spinal cord occurs between the thecal sac and the spinal cord itself. Dentate ligaments and arachnoid structures anchor the spinal cord within the thecal sac, allowing it to maintain its dorsoventral orientation and an approximate craniocaudal position, withstanding all the movements of the bony spinal column. This, in turn, dampens the effects that such movements would have on the neuronal and vascular elements within the cisterna magna.

The critical neurophysiological role of the intra-and extradural suspension of the cisterna magna structures in the integrity of the spinal cord movement and motion has not been clinically acknowledged. However, motion of the suspended neural structures at this location have been studied extensively. For example, human ultrasound and phase-contrast magnetic resonance studies (PCMR) demonstrate that the cerebellum, the tonsils, and spinal cord pulse in synchrony with the cardiac and respiratory cycles, traveling caudally in systole and rostrally in diastole, as well as moving in various planes in response to other external forces [1,3,4,8,9,10,11].

### 1.2. Cranio-Spinal Suspension at the CM in Chiari Malformation Type I

In Chiari Malformation Type I (CM1), however, low-lying tonsils obstruct the cisterna magna at the foramen magnum, crowding out this subarachnoid space around the cervicomedullary junction, sometimes related to a small posterior fossa. This compromises an essential juncture between the cranial and spinal compartments, inhibiting craniocaudal CSF flow in response to instances of increased intracranial pressure and causing compression of the surrounding neural structures [10]. This may lead to clinical symptoms of CMI such as tussive or Valsalva-induced suboccipital headaches, and sensory or motor disturbances, sometimes also in the context of a syrinx. Additional symptoms may include blurry vision, nystagmus, hearing loss and tinnitus, imbalance, and dysfunction of the lower cranial nerves or even the brainstem [12,13]. Given the underlying obstructive flow pathology, one primary aim of Chiari surgery is to restore free CSF transit via functional restoration of the cisterna magna with posterior fossa decompression (PFD), with or without duraplasty [10,13,14,15,16].

### 1.3. Viscoelastic Suspension Structures in CMI

Despite the understanding of tonsillar herniation and these resultant effects on CSF dynamics, many patients with radiographic tonsillar herniation are asymptomatic, and our understanding of what ultimately leads to symptomaticity continues to evolve. Although the obstructive and hydrodynamic pathophysiology of Chiari in the cisterna magna is well-studied, less understood are the implications and effects of this pathology on the aforementioned suspension structures and their role in the neurophysiology of the craniocervical junction. In patients with CMI, this complex continues to suspend the dura, cerebellum, cervicomedullary junction, and spinal cord within the cisterna magna, in its usual role as a shock-absorber and motion-adapter. However, dynamic imaging modalities have demonstrated pathologically increased spinal cord and tonsillar motion in symptomatic pre-operative CM1 patients compared to healthy controls and post-operative CM1 patients [11,17,18]. In the context of a crowded posterior fossa and obstructed cisterna magna with abnormal CSF flow, such non-physiologic forces are constantly exerted on the suspended cerebellomedullary structures, including anatomic distortions and mass effect from tonsillar herniation, so-called “tonsilar pistoning” [18], and altered CSF pulsations, which continue to occur with the normal and constant cardiac and respiratory cycles, and mechanical movements of the head and neck. As valuable as preoperative dynamic imaging modalities may be, none are more revealing of CM1 pathology than intraoperative surgical findings [15,19].

In this descriptive study, we aim to illustrate a novel concept of compromised suspension structures at the craniocervical junction in CM1 using intraoperative images and videos to support the previously observed imaging findings [1,3,4,8,9,10,11]. Based on the observed findings, we propose the possibility that within CMI pathology, a disrupted complex of intra- and extradural suspensive structures is rendered less able to absorb the physiologic and pathologic shocks occurring at the cisterna magna. This in turn may contribute to a form of “tethering” of the cervicomedullary tissues, which may also play a previously unstudied role in the pathophysiology of CM1.

## 2. Materials and Methods

This is a descriptive study aiming to illustrate a novel concept through intraoperative images of soft tissue Chiari pathology derived from a single-surgeon case series of intraoperative images and videos. After obtaining IRB approval (RIH IRB #1579251) we identified patients with CM1 who underwent Chiari surgery at a single institution by a single surgeon during a 10-year period (2011–2021). We reviewed intraoperative images and videos taken during these cases to select examples of salient suspensive structures as introduced above.

By way of clinical background, the CMI patients whose intraoperative images are shown here were considered surgical candidates if they (1) complained of Chiari-pattern headaches (increasing with Valsalva or exertion), as well as any other disabling symptoms consistent with CMI such as motor or sensory disturbances; and (2) demonstrated radiographic evidence of tonsillar herniation below the foramen magnum, with or without a syrinx. All surgical procedures included PFD (approximately a 2.5 cm × 4 cm suboccipital craniectomy or similar determined by the patient’s occipital bone morphology, with C1 laminectomy), dural opening and microsurgical arachnoid resection and exploration of the posterior fossa including the obex and basolateral cisterns, with partial tonsillar cauterization only if the tonsils appeared gliotic or particularly obstructive. Duraplasty was performed using a AlloDerm patch, 5-0 Prolene suture, and a Tisseal and Duragen overlay (Figure 2). Table 1 summarizes the clinical details of the patients represented in the images. All patients received a full work up with brain and entire spine MRI during their preoperative work-up.

## 3. Results: Illustrations, Descriptions and Intraoperative Findings of Spinal Cord Movement and Suspension

### 3.1. Movement of the Spinal Cord in Healthy Subjects

In Figure 3, MRI (a,b) and ultrasound (c,d) are shown of a healthy (non-CM1) control subject, illustrating the craniocervical junction in flexion (a,c) and extension (b,d). Intradural suspension of the spinal cord is at play helping it maintain its dorsoventral position. The subarachnoid space is dynamic between these two movements, as evidenced by the changing distance between the yellow “+” markers in Figure 3c (0.48 cm) and Figure 3d (0.55 cm). The C1 nerve roots (red asterisks) are also shown, as additional suspension structures which restrict the movement of the thecal sac and spinal cord in space at the craniocervical junction.

### 3.2. Myodural Bridges

The MDBs are a thickened soft tissue fibrous structure that extend from suboccipital musculature (rectus capitus minor, rectus capitus major, and obliquus capitus inferior muscles), traversing through the atlanto-occipital and atlanto-axial interspaces and merging into the superficial dura (Figure 1). MDBs bridge these muscles to the dura as a transitional structure that microscopically become the superficial layer of dura [1,3,5,6,20]. These are always exposed and resected in Chiari surgery, both with and without duraplasty, as their mass further constricts the epidural space around the cisterna magna. Figure 1 is a posterolateral illustration of the craniocervical junction with magnified sagittal and axial views. The myodural bridges are seen connecting the cisterna magna dura to the suboccipital muscles (asterisks).

In Figure 4, views of the suboccipital region following muscle dissection and removal of the posterior arch of C1 are shown. The myodural bridges can be seen just inferior to the inferior occiput prior to (Figure 4a) and following removal of the inferior occiput (Figure 4b). As the myodural bridges are lifted with a Gerald forcep (Figure 4c), the dural attachment is clearly visualized and manipulation of the myodural bridges leads to a corresponding manipulation of the dura (Figure 4d), demonstrating their suspensive role. This dynamic interplay between the myodural bridges and the underlying dura are further illustrated in Appendix A. In this video, the myodural bridges are demonstrated pulling the dura, demonstrating their full integration into the superficial layer of dura and their role in mobilizing the dura.

### 3.3. Pathological Arachnoid Structures

In CMI patients treated with PFD with duraplasty, various arachnoid pathologies have been reported [15,19] and observed by our group as well, including but not limited to arachnoid adhesions, bands, thickening, and opacifications. In Figure 5, after tacking up the dura, the underlying arachnoid can be visualized and is subsequently resected (Figure 5a). The arachnoid is thickened in these areas and the underlying tonsils appear gliotic, with numerous associated arachnoid adhesions that appear to tether the cerebellar tonsils to the arachnoid and overlying dura (Figure 5b). These adhesions are carefully dissected out and cut (not shown). In Appendix A, arachnoid bands are seen not only obstructing CSF flow, but also exerting direct traction on the tonsils and cervicomedullary junction which are seen functionally suspending these structures within the cisterna magna. In Appendix A, an intraoperative pre-dural opening ultrasound shows a small dorsal arachnoid adhesion or web (upper left of video), which appears to tether the spinal cord to the dorsal dura. This band is seen tugging and moving the spinal cord with the impulses of systole, diastole, inspiration, and expiration, as well as direct intraoperative manipulation of the bands themselves.

### 3.4. Tonsils

In Figure 5 (above), gliotic tonsils are shown, with whitening or graying pial areas. In Figure 6 (below), multiple arachnoid adhesions can be seen tethering the cerebellar tonsils to each other and to the spinal cord. Following cauterization of the gliotic inferior cerebellar tonsils, the dentate ligament, deformed by prolonged pressure from the tonsils, can be visualized. In Appendix A, ultrasound image prior to dural opening demonstrates herniated tonsils pistoning against the spinal cord, not only producing a strain on the spinal cord but tethering it to the vertebrobasilar junction, lacking any noticeable movement or CSF pulse-synchronized motion.

### 3.5. Dentate Ligaments

The dentate ligaments are distinct, individual fibrous structures suspending the spinal cord to the inside of the thecal sac dura, extending from the level of the C1 down to T12 vertebrae. They work to maintain a static yet dynamic position of the spinal cord, tethering the spinal cord laterally to preserve its dorsoventral and rotational orientation as well as an approximate craniocaudal position. For this reason, they are sometimes cut during intradural spinal surgery, for example if the spinal cord rotation or retraction is necessary in order to access more ventral intradural lesions. In Figure 6d, dentate ligaments are shown after cauterization of the tonsils, which had obstructed (Figure 6a) and exerted mass effect on the dentate ligament itself. The dentate ligament is seen to be deformed and impinged by prolonged pressure from the tonsils.

## 4. Discussion

It is well-accepted that CMI involves symptoms associated with obstructed CSF flow at the cisterna magna due to tonsillar herniation, often in a small posterior fossa. However, many patients with radiographic tonsillar herniation are asymptomatic, and the degree of herniation and posterior fossa crowding do not always directly correlate to CMI symptoms [1,2,16,21]. This suggests that additional less appreciated pathologic factors not seen on standard imaging may also be at play in addition the obstruction itself leading to symptoms. Here, we explore a potential role that these intra- and extradural suspensive structures may play as a factor in the compromised neurophysiology of CM1—a previously rather unexplored and less acknowledged area.

One factor in this phenomenon is the concept of “occult” spinal biomechanical instability. Frank biomechanical instability is not associated with most cases of CMI without gross craniovertebral junction deformity. However, there is some suggestion that mechanical factors may at least play a role in pathology. Among CMI patients, there is an increased incidence of connective tissue disorders such as Ehlers Danlos Syndrome (EDS) [22], in which the myodural bridges demonstrate an altered collagen superstructure with impaired shock absorption [1]. Among patients with previously asymptomatic Chiari, some have been described to exhibit biomechanically triggered symptoms such as following head and neck trauma or whiplash [21,23,24]. Moreover, anatomical studies have shown distorted atlanto-occipital and atlanto-axial joints in CMI [16]. Some authors have even shown cases with normal bony anatomy, where mechanical fixation alone of the atlantoaxial joint has helped with Chiari symptoms [25,26].

Moreover, several intradural motion-related dynamics are at play in CMI, as the shrunken-to-obliterated subarachnoid space at the foramen magnum in CMI has been shown to change with neck flexion and extension [2,8,14]. Subsequently increased tonsillar tissue strain and motion have been demonstrated in CM1 patients, as well as an association with increased tonsillar tissue strain and Valsalva headaches, with reduction after PFD [14,17,18]. Similar findings are true for spinal cord motion, with greater spinal cord motion in CM1 compared to controls, resolved after PFD [11,17]. In our previous collaborative discussion on “A new hypothesis for the pathophysiology of symptomatic adult Chiari malformation” [2], we suggested that altered CSF dynamics in a crowded posterior fossa may exert additional stress on epidural and extradural suspension structures, resulting in their repeated activation as they cushion the strain on the cerebellum and cervicomedullary junction. Given that the motion and suspension of neural tissue is associated with symptoms and their resolution, craniospinal suspension may be a link to symptomatic CM1.

Our intraoperative images support the notion that this may be a secondary biomechanical pathology within CM1. We demonstrate myodural bridges, arachnoid bands, and dentate ligaments in intraoperative videos and images of PFD for CMI, demonstrating pathologic spinal cord and tonsillar motion linked to their anatomical distortion. These distorted viscoelastic suspensive elements at the craniocervical junction are seen interplaying with the primary pathology of altered CSF dynamics. These structures are shown acting to suspend the cerebellum, tonsils, and cervicomedullary junction within the obstructed cisterna magna, and becoming structurally and functionally compromised within this environment, at times even causing more obstruction themselves. We hypothesize a vicious cycle of altered CSF dynamics and cisterna magna obstruction interacting with disrupted suspensive elements at the craniocervical junction, leading to spinal cord tethering, an un-studied pathophysiology in CM1.

Recently, active stretch-sensitive and nociceptive nerve endings as well as impaired viscoelasticity were shown in filum terminale specimens from hypermobile EDS (hEDS) patients undergoing surgery for tethered cord syndrome—demonstrating tendon-like properties in the filum terminale, which becomes overworked in collagen disease and then fails to dampen stretch injuries to the conus medullaris and spinal cord [27]. Similarly, myodural bridges have been shown to have proprioceptive properties and histological examinations have shown nervous structures within them, with some describing a myo-reflexive response involving dural nociception and suboccipital muscle contraction mediated by the myodural bridges [28]. Given the role of the myodural bridge as both a passive dural anchor and possibly an active stabilizer of the cisterna magna and its contents, as well as the other structures shown here in our paper, perhaps one area of further study could be to explore whether a form of “tethering” may herein be at play at top of the spine in CMI—the craniospinal junction—just as it is at the bottom of the spine in hEDS tethered cord—the filum terminale. Future studies which may be useful in studying this phenomenon further may include imaging tissue motion and suspension, such as through finite analysis modeling.

## 5. Conclusions

Dysfunctional myodural bridges, pathologic arachnoid structures, and impinged dentate ligaments are viscoelastic suspension elements of the cerebellomedullary junction shown to be disrupted in our intraoperative images and videos during PFD for CMI. We propose that such pathology may interplay or result from the primary CMI pathology of cisterna magna. This novel hypothesis that these structures and their inelastic failure may contribute to CM1 pathophysiology may be a ripe area for continued study in CMI.

## Figures and Tables

**Figure 1 jcm-11-07437-f001:**
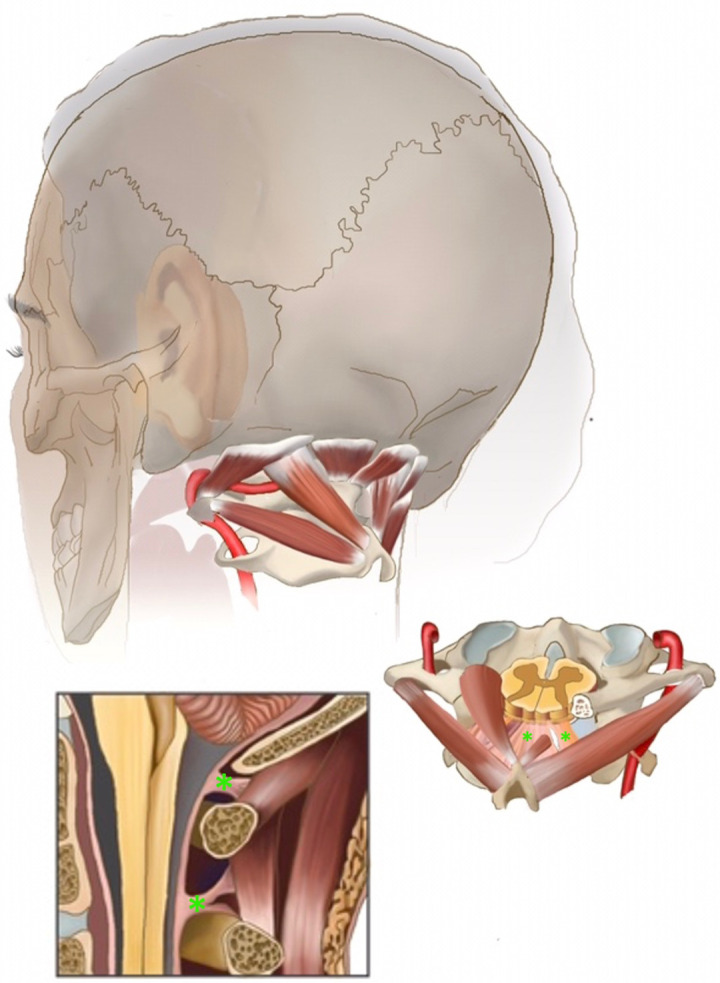
The suspensive myodural bridge complex at the cisterna magna. Posterolateral illustration of the craniocervical junction with magnified sagittal and axial views. The myodural bridges are seen connecting the cisterna magna dura to the suboccipital muscles (asterisk). (Original illustration by Kendall Lane, BFA, Department of Medical Illustration, Warren Albert Medical School, Brown University).

**Figure 2 jcm-11-07437-f002:**
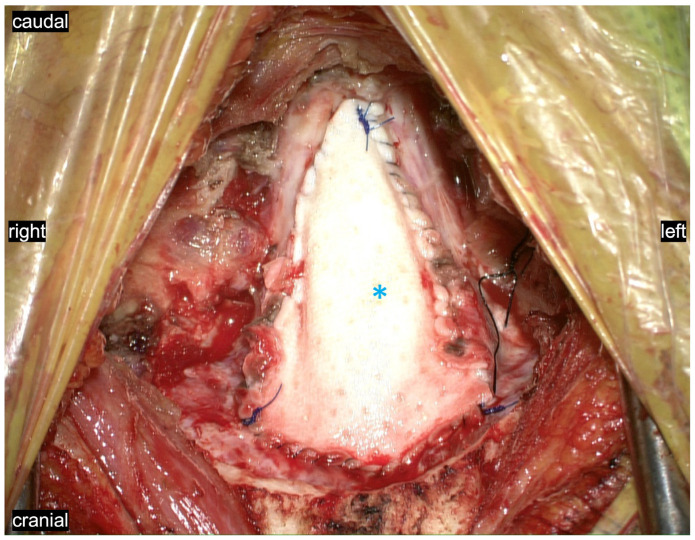
A watertight duraplasty is performed using an AlloDerm patch (asterisk) with a running 5-0 Prolene suture following a standard Y-shaped dural opening. Subsequently, a Duragen onlay and Tisseal are applied to reinforce this dural closure (not shown). This duraplasty pulsates with the cardiac and respiratory cycles as shown in our included operative videos, synchronizing the flow of CSF between cranial and spinal compartments.

**Figure 3 jcm-11-07437-f003:**
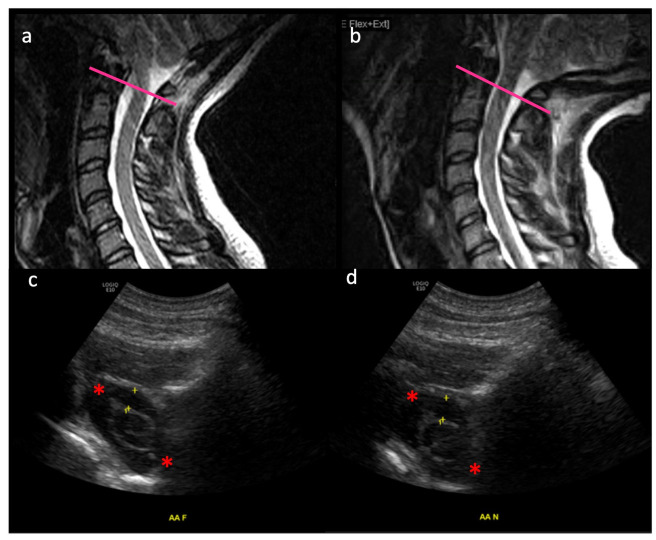
MRI (**a**,**b**) and ultrasound (**c**,**d**) of a healthy (non-CM1 patient) are shown, illustrating the craniocervical junction in flexion (**a**,**c**) and extension (**b**,**d**). Approximate axial level of ultrasound images is denoted by the pink line on MRI images. Intradural suspension of the spinal cord is at play helping it maintain its dorsoventral position. The subarachnoid space varies between these two movements, as evidenced by the distance between the yellow “+” markers in 2c (0.48 cm) and 2d (0.55 cm). The C1 nerve roots (red asterisks) are also shown, as additional suspension structures which restrict the movement of the thecal sac and spinal cord in space at the craniocervical junction. These are studies from a healthy volunteer. This study [1,5] was conducted in 2021 under the RIH IRB 1338182 and supported by a seed Grant sponsored by the Rhode Island Medical Imaging Department.

**Figure 4 jcm-11-07437-f004:**
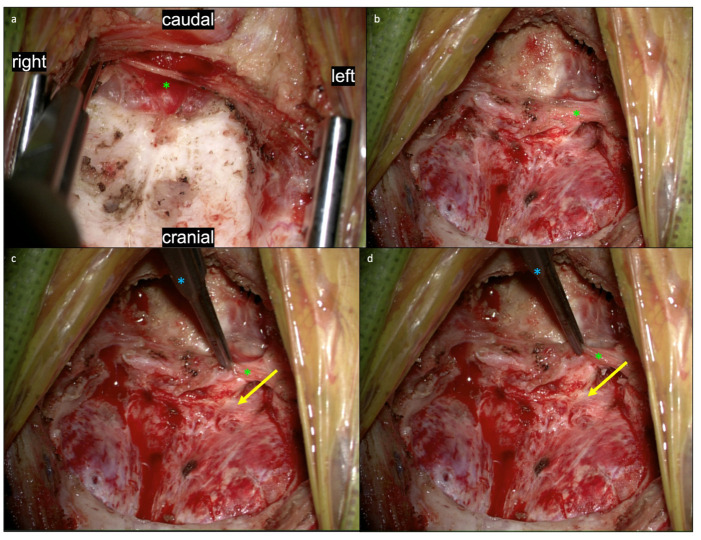
Views of the suboccipital region following muscle dissection and removal of the posterior arch of C1. The myodural bridges (green asterisk) can be seen just inferior to the inferior occiput prior to (**a**) and following (**b**) removal of the inferior occiput. As the myodural bridges are lifted with a Gerald’s forcep (blue asterisk) (**c**), the dural attachment (yellow arrow) is clearly visualized and manipulation of the myodural bridges leads to a corresponding manipulation of the dura (**d**), demonstrating their suspensive role. This dynamic interplay between the myodural bridges and the underlying dura are further illustrated in Appendix A.

**Figure 5 jcm-11-07437-f005:**
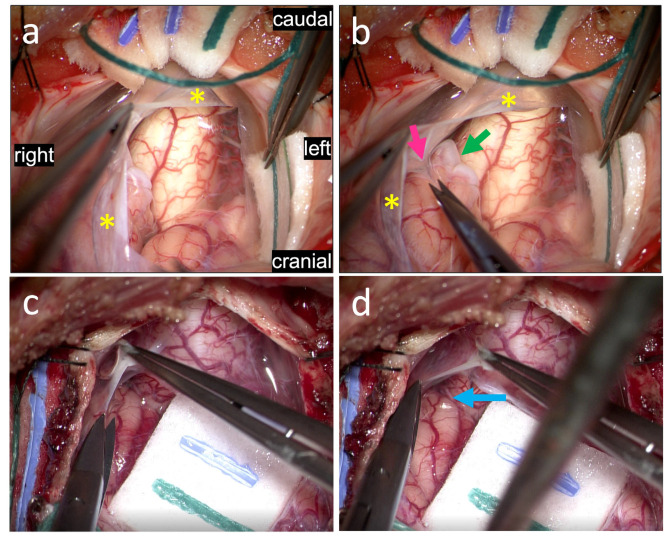
View of the posterior craniocervical junction following durotomy. After flapping up the dura, the underlying arachnoid (yellow asterisk) can be visualized and is subsequently resected (**a**). The arachnoid is thickened in these areas and the underlying tonsils appear gliotic (green arrow), with numerous associated arachnoid adhesions (magenta arrow) that appear to tether the cerebellar tonsils to the arachnoid and overlying dura (**b**). These adhesions are carefully dissected out and cut (not shown). For comparison, a non-CM1 case is shown, with non-gliotic tonsils free from arachnoid tethering (**c**,**d**).

**Figure 6 jcm-11-07437-f006:**
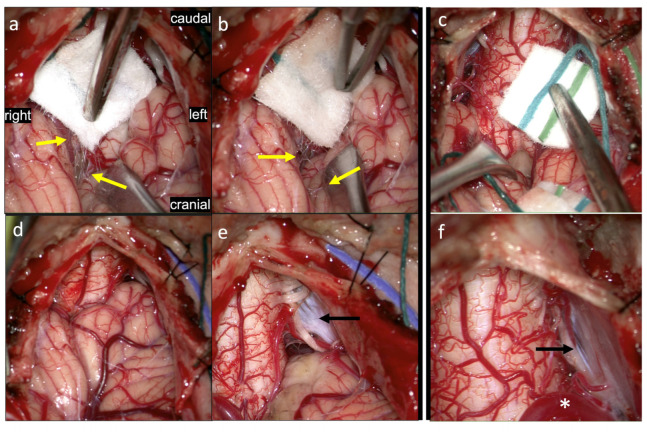
View of the posterior craniocervical junction in CM1 following resection of the overlying arachnoid membrane. Multiple arachnoid adhesions (yellow arrows) can be seen tethering the cerebellar tonsils to each other (**a**) and to the spinal cord (**b**). For comparison, a non-CM1 case is shown, with a view of the obex and the absence of tethering by arachnoid adhesions (**c**). Back to the CM1 case, obstructive inferior cerebellar tonsils (**d**) are then cauterized (**e**), revealing a dentate ligament (black arrow) deformed by prolonged pressure from the tonsils. Comparison is made to the same non-CM1 case (**f**), in a zoomed-in and superolaterally angled view, with a normal-appearing dentate ligament seen between the lower cranial nerves (black arrow). Also notable is as a vermian PICA loop (asterisk) visible here given the lack of tonsillar herniation in this non-CM1 patient.

**Table 1 jcm-11-07437-t001:** Clinical details of selected patients.

Figure/ Video	Age, Sex	Pre-Operative Symptoms	Duration of Symptoms	Preop MRI	Syrinx Y/N	Post-Operative Follow-Up
Figure 2	47F	Valsalva induced headaches, blurry vision and episodic tinnitus and hearing loss, cognitive slowing and brain fog	>3 months	3.8 mm tonsillar herniation	N	Complete resolution of preopeartive symptoms (3 m post surgery)
Figure 4, Figure 5, Appendix A	10F	Mix of suboccipital headaches and “migraine” type frontal headaches refractory to medical management and physical therapy, light sensitivity	1 year	7.5 mm tonsillar herniation	N	Gradual resolution of preoperative headaches over the course of 1year post surgery
Figure 5	47F	Valsalva induced headaches, neck pain, episodic vergigo and dizzines, difficulty swallowing, fatigue and cognitive slowing with brain fog, co-morbiditie of hypermobile Ehlers-Danlos-Syndrome and tethered cord syndrome	3 years	5.8 mm tonsillar herniation	Y	Headaches and preoperative dizziness gradually improved, residul back and leg pain required tethered cord release 6 months post Chiari decompression, stable syrinx
Appendix A	35F	Suboccipital and frontal headaches with co-morbdity of known left sided migraines refractory to medical management, episodic vertigo/dizziness	3 years	6 mm tonsillar herniation	N	Resolution of headaches including improved frequency and intensity of left sided mgraines 1y post surgery
Appendix A	27F	Valsalva induced headaches and neck pain, subjective weakness in the right hand, brain fog, episodic vertigo/dizziness with drop attacks, brain fog, difficulties with proprioception	>3 years	22 mm tonsillar herniation, reduced clivoaxial angle in supine position <125) no radiographic evidence of craniocervical instability	Y	Residual neck pain and neck spasm, occasional headaches and vertigo, resolution of the remainder symptoms at the 1y post surgery, near complete resolution of the syrinx
Appendix A	6M	Global headaches with “seizure-like” episodes, seizures ruled out, neck spasms, inconsolible and aggressive behaviour during pain episodes, symptoms refractory to intensive medical and physical therapy	>12 months	11 mm tonsillar herniation	N	Significantly improved pain and symptom episodes compared ot preop 1 months post surgery sustained at 6 months post decompression
Figure 5c,d; Figure 6c,f (non-Chiari “control”)	11M	progressive papilledema, visual acuity decline, scoliosis	4 years	absent tonsillar herniation. progressive thoracic syrinx and scoliosis	Y	Continuosly Improved papilledema and visual acuity per ophthalmological follow up assessment in the year following decompression, stable syrinx

## Data Availability

Not applicable.

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
