# Peer review of "Compromised Cranio-Spinal Suspension in Chiari Malformation Type 1: A Potential Role as Secondary Pathophysiology"

_jcm, 2022, doi:10.3390/jcm11247437_

Round 1

Reviewer 1 Report

The authors present an interesting manuscript regarding an understudied condition in CM1 which is the role of craniospinal suspension. Their laxity, as it can happen in some collagenopathies, can contribute to symptoms and their "tethering", as suggested by the authors, could be considered as a surgical phase.

The draft is well presented with high quality of pictures by means of schematic figures and Intraoperative images.

Author Response

We appreciate Reviewer 1’s review of our manuscript.  We did not note any specific comments made for particular revisions from this reviewer requiring our response.  However we have updated the manuscript based on Reviewer #2's comments and they do address the domains that Reviewer #1 has checked off as being areas of improvement.  The response to Reviewer #2 is also attached in case it is helpful for Reviewer #1.

Reviewer 2 Report

The authors  described anatomical and physiological suspension for brain stem and cerebellum, which means that  dysfunctional myodural bridge, arachnoid structures, and impinged dentate ligaments were viscoelastic suspension elements of the cerebello-medullary junction. This new idea which suggests pathophysiology of CSF dynamics at carniocervical junction in Chiari malformation type (CMI), will be much interesting for surgeons, neurologist and researcher of CMI. Many intra-operative photos and Video are very impressive  but the authors should indicate the objective data as scientific article, for example using echogram or PCMR, velocity, distance or any other strategy.

I recommend that this manuscript should be considered as the cases report etc. because this manuscript does not include the scientific evaluation and data, if the authors are unable to show the objective data as scientific article, for example using echogram or PCMR, velocity, distance or any other strategy.

There are some suggestion. In this manuscript, Figures are very important, the assertion of authors are shown in Figures, so that Figures must be clearly understood by the readers.

1: Figure 2, 3, 4,5 and 6: please put into each photos the marks which show  legends.

2: Figure 3, 4, 5 and 6: please demonstrate the orientation of the direction (right/left, crania/caudal).

3: Figure 3,4,5, and 6: suggest the comparison between other diseases (not CMI, for example tumor etc.) and CMI cases.

4: Figure 3: the authors should also demonstrate the data at the craniocervical junction.

Round 2

Reviewer 2 Report

The authors properly revised the manuscript, according to reviewer's comments and they explained the ideas at detail. I am interested and agree with the ideas by the authors. Photos which are demonstrated in the manuscript are the great value as information for surgeons and researchers. As the author mentioned, I recommend that this manuscript should be re-categorized and published in Chiari and syringomyelia special issue, for example "special review""special case report" etc, because there are a few case and scientific evidence.

Author Response

Thank you. We welcome the re-classification of this paper as the reviewers wish, as discussed in the last round of comments.